# Enhancing *DLG2* Implications in Neuropsychiatric Disorders: Analysis of a Cohort of Eight Patients with 11q14.1 Imbalances

**DOI:** 10.3390/genes13050859

**Published:** 2022-05-12

**Authors:** Veronica Bertini, Roberta Milone, Paola Cristofani, Francesca Cambi, Chiara Bosetti, Filippo Barbieri, Silvano Bertelloni, Giovanni Cioni, Angelo Valetto, Roberta Battini

**Affiliations:** 1Cytogenetic Unit, Department of Laboratory Medicine, Azienda Ospedaliero-Univeristaria Pisana, Via Roma 57, 56100 Pisa, Italy; v.bertini@ao-pisa.toscana.it (V.B.); f.cambi@ao-pisa.toscana.it (F.C.); 2Department of Developmental Neuroscience, IRCCS Fondazione Stella Maris, 56125 Pisa, Italy; rmilone@fsm.unipi.it (R.M.); paola.cristofani@fsm.unipi.it (P.C.); chiara.bosetti@fsm.unipi.it (C.B.); giovanni.cioni@fsm.unipi.it (G.C.); rbattini@fsm.unipi.it (R.B.); 3Mental Health Department, ASL Toscana Nordovest, 56100 Pisa, Italy; filippo.barbieri@uslnordovest.toscana.it; 4Pediatric Endocrinology, Department of Obstetrics, Gynecology and Pediatrics, Azienda Ospedaliero-Universitaria Pisana, Via Roma 57, 56100 Pisa, Italy; s.bertelloni@ao-pisa.toscana.it; 5Department of Clinical and Experimental Medicine, University of Pisa, 56100 Pisa, Italy

**Keywords:** array-CGH, neurodevelopmental disorders, intellectual disability, ADHD, autism spectrum disorder, protein-coding transcripts

## Abstract

Neurodevelopmental disorders (NDDs) are considered synaptopathies, as they are due to anomalies in neuronal connectivity during development. *DLG2* is a gene involved insynaptic function; the phenotypic effect of itsalterations in NDDs has been underestimated since few cases have been thoroughly described.We report on eight patients with 11q14.1 imbalances involving *DLG2*, underlining its potential effects on clinical presentation and its contribution to NDD comorbidity by accurate neuropsychiatric data collection. *DLG2* is a very large gene in 11q14.1, extending over 2.172 Mb, with alternative splicing that gives rise to numerous isoforms differentially expressed in brain tissues. A thorough bioinformatic analysis of the altered transcripts was conducted for each patient. The different expression profiles of the isoforms of this gene and their influence on the excitatory–inhibitory balance in crucial brain structures could contribute to the phenotypic variability related to *DLG2* alterations. Further studies on patients would be helpful to enrich clinical and neurodevelopmental findings and elucidate the molecular mechanisms subtended to NDDs.

## 1. Introduction 

Neurodevelopmental disorders (NDDs) such as intellectual disability (ID), autism spectrum disorder (ASD), attention deficit hyperactivity disorder (ADHD), epilepsy, and atypical cognitive or behavior features, are considered synaptopathies (i.e., provoked by synaptic function abnormalities) [1,2,3,4]. The precise patterns of neuronal assembly and connectivity during development are fundamental for neuromodulation and have been increasingly studied in NDDs [5]. Mutations and copy number variations (CNVs), inherited or de novo, in genes encoding for cell adhesion molecules (e.g., for neurexin–neuroligin, such as *NRXN1–3* or *NLGN1–4*) and scaffolding proteins (e.g., discs large MAGUK scaffold proteins or Shank 1–3) or for modulating synaptic gene transcription, protein synthesis and degradation, and synaptic elimination (e.g., *FMR1*, *TSC1–2*, *MECP2*, and *UBE3A*) have been related to NDDs, since they interfere with synapse structure and function [2]. Both ‘gain-of-function’ and ‘loss-of-function’ mechanisms should be considered at the basis of synaptopathies, due to provoking imbalanced synaptic functioning [4]. 

Among genes involved in synapse formation, discs large MAGUK scaffold protein 2 (*DLG2*) has recently received attention. *DLG2 *(*603583; ENSG00000150672) is a very large gene in 11q14.1, extending over 2.172 Mb, with numerous alternative transcripts (https://www.ensembl.org on 5 April 2022). Its products are scaffolding postsynaptic density protein-93 (PSD-93) or channel-associated protein of synapses-110 (Chapsyn-110) [6], which acts as a postsynaptic multimeric scaffold protein belonging to the membrane-associated guanylate kinase (MAGUK) family. It binds to both cytoskeleton proteins and signaling complexes, influencing the development, plasticity, and stability of synapses [7]. It is involved in signaling and in the molecular organization of multiprotein complexes in postsynaptic density [6], and it is distributed in glutamatergic excitatory brain synapses [7]. In particular, it is recruited in NMDA receptors, regulating their surface expression and interacting with their cytoplasmic tails (www.genecards.org, accessed on 5 April 2022), and inwardly rectifying potassium channel (Kir) clusters [8]. All these data have arisen from electron microscopy studies, in vitro analyses, and experiments in mice models.

In humans, copy number variations (CNVs) involving *DLG2* were first associated with schizophrenia [9,10] and, subsequently, with NDDs, including ASD, ID, and bipolar disorder [11,12,13,14]. Epilepsy has been rarely reported [15,16]. In most cases, the clinical descriptions are very scanty, as patients are part of large cohorts for case-control analyses in which recurrent CNVs in association with specific neuropsychiatric disorders have been investigated.

Patients with NDDs and *DLG2* alterations are reported in Decipher (https://www.deciphergenomics.org/, accessed on 5 April 2022), but the phenotypic features are limited also in these cases.

Here, we report on a cohort of eight patients with neuropsychiatric features and CNVs in 11q14.1 that include *DLG2* in order to provide a more extensive clinical characterization. 

## 2. Materials and Methods

### 2.1. Patients

The 8 patients were recruited during a 10-year period at the IRCCS Stella Maris Foundation and at AOU Pisana. They presented neurodevelopmental and neuropsychiatric disorders that deserved follow-up after diagnosis. Patients with CNVs in the 11q14.1 region encompassing *DLG2*, either isolated or associated with other CNVs, were selected for this study.

All the patients underwent a detailed neuropsychiatric and psychological standardized observation. Brain MRI was acquired for 7/8 patients and EEG polygraphy was recorded in 3/8.

### 2.2. Molecular Analysis 

Array comparative genomic hybridization (array-CGH) was performed in all 8 patients. The genomic DNA of the patients was isolated from peripheral blood by standard methods; DNA from healthy subjects (one male and one female) was used as the control (Agilent Technologies, Santa Clara, CA, USA). A total of 200 ng of genomic DNA, both from each patient (test sample) and the control (reference sample), was differentially labelled with Cy5-dCTP or with Cy3-dCTP using random primer labelling according to the manufacturer’s protocol (Agilent). The labelling reactions were applied to the 60K oligoarrays (Agilent, Santa Clara, CA, USA) and incubated for 24 h at 65 °C in an oven. The slides were washed and scanned using an Agilent scanner, and the identification of individual spots on the scanned arrays and quality slide evaluation were performed with Agilent-dedicated software (Feature Extraction, Agilent, Santa Clara, CA, USA). The 60K slides had a 41 Kb overall median probe spacing (33 KB in RefSeq genes).

The copy number variants (CNVs) were identified with Cytogenomics 3.0.6.6. (Agilent, Santa Clara, CA, USA) using an ADM-2 (aberration detection method-2) algorithm. This algorithm identifies aberrant intervals in a sample that has consistently high or low log ratios based on statistical score. ADM-2 uses an iterative procedure to find all genomic intervals with a score above a user-specified statistical threshold value. The threshold was set to a minimum of 6, with the minimum number of probes required in a region of 3 and a minimum absolute log ratio of 0.25. The score represented the deviation of the weighted average of the normalized log ratios from its expected value of zero and incorporated quality information about each probe measurement.

CNVs classified as pathogenic, likely pathogenic, or variant of unknown significance (VUS) were reported according to the guidelines of the Italian Society of Human Genetics (https://www.sigu.net, accessed on 5 April 2022), the American College of Medical Genetics guidelines [17], and the European Guidelines for constitutional cytogenetic analysis [18].The CNV classification was performed using databases such as the Database of Genomic Variants (DGV) (http://projects.tcag.ca/variation, accessed on 5 April 2022), the Database of Chromosome Imbalance and Phenotype in Humans using Ensembl Resources (Decipher) (https://www.deciphergenomics.org/, accessed on 5 April 2022), and the University of California Santa Cruz (UCSC) Genome Browser (https://genome.ucsc.edu/, accessed on 5 April 2022).

Pubmed (http://www.ncbi.nlm.nih.gov/pubmed, accessed on 5 April 2022), Online Mendelian Inheritance in Man (OMIM) (http://www.omim.org, accessed on 5 April 2022), and GeneCards (https://www.genecards.org, accessed on 5 April 2022) were also consulted for evaluating genotype–phenotype association.

The pathogenic CNVs detected in the 8 patients were reported according to the Genome Reference Consortium Human Build 38 (GRCh38/hg38). The segregation analyses in the parental DNA were performed by qPCR or by array-CGH.

The transcripts were researched using Ensembl (https://www.ensembl.org, accessed on 5 April 2022); the data on the expression profiles were researched using GTEx (https://www.gtexportal.org/home, accessed on 5 April 2022), and UCSC (https://genome.ucsc.edu/, accessed on 5 April 2022).

## 3. Results

### 3.1. Description of the Patient Cohort

The age of the patients ranged from 4 years old to 16 years old. Seven patients were male, and one was female. The neuropsychiatric and phenotypic features of the eight patients are summarized in Table 1 and Appendix A.

### 3.2. CNVs Detected in the Patient Cohort

The 11q14.1 CNVs detected by array-CGHranged from 68 Kb to 1.27 Mb; they included one intragenic duplication of *DLG2* (Pt.8), six intragenic deletions (Pt.2–7), two patients (Pt.6 and 7) sharing an identical genomic imbalance, and one larger deletion (Pt.1) harboring additional genes at the 5’ of *DLG2*; two cases (Pt.3 and 6) presented additional rare CNVs (Table 2).

The pattern of inheritance was assessed in six patients: in three cases, the CNVs were inherited from the mother and, in three, from the father; the segregation analysis was not available for two patients (Table 2).

The biological roles of genes included in these CNVs are summarized in the Appendix A.

### 3.3. DLG2 Transcripts and Expression

*DLG2* extends over 2.172 Mb and is encoded on the reverse strand. According to Ensembl, *DLG2* (ENSG00000150672) gives rise to 32 transcripts. All the protein-coding transcripts that overlapped with the CNVs of our patients were selected, and for each of them, the exons involved in these imbalances were exactly localized (Figure 1; Table 2; Appendix A).

These data highlight that all the CNVs detected in our patients altered the nucleotide sequence of one or more transcripts; also, the duplication of patient 8 gave rise to a transcript that harbored two copies of the same exon, modifying the amino acid sequence of the protein.

The main features of the 13 transcripts selected are summarized in Appendix A; DLG2-321 and DLG2-221 are the same transcripts described by Reggiani et al., 2017 [12]. 

The expression of the transcripts across tissues in GTEx, UCSC, and in [12] was also evaluated. These transcripts showed very different expression profiles: some of them were scarcely expressed (e.g., DLG2-218, DLG2-202, DLG2-205, and DLG2-230), others were ubiquitous (e.g., DLG2-203 and DLG2-215), and several (i.e., DLG2-207, DLG2-201, DLG2-220, DLG2-229, and DLG2-221) showed a high and prevalent expression in the central nervous system (CNS). Some expression profiles are exemplified in Figure 2, while the rest are visualized in Appendix A.

## 4. Discussion 

This paper reported on eight patients with CNVs involving *DLG2* and highlighted the association between this gene and NDDs.

In this cohort, ID appeared as a leitmotif since it was diagnosed in the totality of our patients. In addition to ID, the most common NDDs were ADHD (five out of eight patients), verbal-motor dyspraxia (two patients, as well as one reported with motor clumsiness), and receptive-expressive language disorder (two patients). One patient presented focal epilepsy, and the other three showed EEG anomalies. In addition, one patient obtained a diagnosis of ASD, and autistic traits were described in two other cases (Table 1).

*DLG2* is a very large gene in 11q14.1, extending over 2.172 Mb, with more than 40 exons that give rise to 32 transcripts. Among these patients, seven had intragenic alterations of *DLG2*, whereas one had a larger deletion harboring additional genes.

In humans, the role of this gene in NDDs has been underestimated, probably due to the fact that *DLG2* shows incomplete penetrance; alterations of this gene have been reported also in healthy people (DGV database), and some patients with NDDs inherited the CNVs from an asymptomatic or paucisymptomatic parent [12], such as six cases in our cohort.

Furthermore, DLG2-203, the MANE transcript (matched annotation from NCBI and EMBL-EBI), seems not to be the most representative transcript for the synaptopathies since it was ubiquitous and was not particularly expressed in the CNS (Figure 2). Additionally, several patients with NDDs reported both in Decipher and in the literature had only intronic deletions in DLG2-203. We added two new cases whose imbalances did not affect the coding sequence of this transcript.

In view of this, all the protein-coding transcripts that overlapped with the CNVs in this cohort were evaluated. Among these transcripts, we focused on those with altered nucleotide sequences with one or more exons harbored in the imbalanced regions (Figure 1; Table 2). 

Since the expression of alternative transcripts is highly tissue-specific [19], we analyzed the expression profiles of the altered transcripts with bioinformatic tools, and interestingly, all the patients had at least one transcript that was highly expressed in the CNS (Figure 2 and Appendix A). 

Three patients had an altered dosage of additional genes in addition to*DLG2* that influenced their phenotype. Among the genes harbored in the deletion of patient 1, *PICALM* deserves attention due to its role in cognition and synaptic plasticity [20]. The two duplications of patient 3in 1q42.2 and 5q14.1 contained several genes. Among them, *SLC35F3* and *JMY* are brain-enriched genes [21]. The former is expressed in excitatory–glutamatergic and inhibitory–γ-aminobutyric acidergic neurons, while the latter is crucial during CNS development, especially in myelination [22,23]. Nothing is known about the phenotypic effects of their duplications. Correlation between the Xp22.31 duplication (patient 6) and NDDs is debated [24] (Appendix A).

An almost constant feature of NDDs is their comorbidity, which can originate complex neuropsychiatric pictures characterized by the concomitance of different disorders [25]. Comorbidity was also evident in this cohort, carrying alterations of *DLG2* and strengthening the idea that the same molecular mechanism could be at the basis of NDDs.

A strict association between ASD and ADHD was evidenced [26], and *DLG2* may concur with both disorders [13,14,27]. Although in our cohort only one patient presented with ASD, ADHD is very frequent, further underlining the overlapping genetic basis between these NDDs.

To the best of our knowledge, epilepsy has been only rarely related to *DLG2* deletion [15,16]; four patients of this cohort showed epilepsy or EEG anomalies. *DLG2* belongs to the gene motif “BGNADP”, which is a network associated with ID and epilepsy, including also *BTD*, *GALNT10*, *NMUR2*, *AUTS2*, and *PTPRD*. Furthermore, *DLG2* has been related to a pathway including *GRIN2B*, and they share an influence on NMDA receptor function [28,29]. 

The direct interaction between the DLG2 and Kirchannels [8,30] determining the functional maturation of spiny projection neurons during a critical period of striatal circuit development [6] may concur with the strong association between epilepsy and ASD [31,32]. It is worth noting that, among patients with epileptiform anomalies, one met the criteria for a diagnosis of ASD, and the other two presented autistic traits.

Psychiatric disorders such as mood disorder, anxiety disorder, obsessive-compulsive disorder (OCD), oppositional-defiant disorder (ODD), and binge eating were present in some of our patients. Mood disorder has been recognized as a neurodevelopmental psychiatric disorder [33]. Shared genetic pathways among synaptopathies including psychiatric disorders have been related to *DLG2* haploinsufficiency both in mice models and in human studies [11,34]. In this study, we focused on a pediatric population in which NDDs were frequent, and we cannot exclude that, following a prospective trajectory until adult age [35], there could be a higher risk of developing psychiatric disorders such as mood disorder, which is especially continuous with ADHD [36].

The interference with the excitatory synaptic transmission in the striatum may explain *DLG2* contribution to such disorders [6,37,38]. The striatum itself is involved in decision making and in executive function with common circuitries in ASD and OCD [39], further corroborating the continuity between NDDs and psychiatric disorders. In addition, reward responsiveness alterations in critical brain regions, including the striatum, have been identified in several psychiatric disorders, such as ADHD, ODD, and binge eating [40,41].

Verbal and motor dyspraxia, clumsiness, and slowness in executive function appeared quite frequently in this population and were signaled also in Decipher patients. These disorders may recognize similar neural substrates due to connectivity anomalies [42] even if, in mice models, the association between *Dlg2* deletion and motor coordination has not been confirmed [34].

The presented data confirmed the association between *DLG2* alterations and multiple NDDs; however, the molecular mechanisms underlying the pleiotropic effect of this gene are not fully understood. Different DLG2 isoforms are deemed to have some functional differences: in addition to not sharing the same exact sequence, they do not have the same pattern of expression in different regions of the CNS. However, according to these data and those available from public databases, it is not yet possible to determine the functional peculiarities of each DLG2 isoform.

With regards to NDDs, functional characterization of the transcripts is even more challenging due to the comorbidity of these conditions. It can be speculated that one of the mechanisms underlying the comorbidity of NDDs is a genetic imbalance of *DLG2*, such as a CNV, that causes the concomitant alteration of several transcripts with different functional roles. Interestingly, magnetic resonance imaging revealed only aspecificanomalies in some of our patients; however, they were not related to the prevalent transcript expression profiles, confirming that NDDs should be considered synaptopathies [1,2,3,4].

Moving from neuropsychiatric to physical features, prenatal and postnatal growth alterations, overweightand obesity, facial dysmorphisms, heart anomalies, and endocrinologic anomalies were other characteristics detected in our cohort of patients (Appendix A) and in patients reported in Decipher. Again, the role of *DLG2* in these clinical features remains to be clarified.

## 5. Conclusions

*DLG2* haploinsufficiency is related to NDDs, with a wide clinical spectrum, ranging from ID, ASD, and ADHD to psychiatric disorders such as mood disorders, anxiety disorders, OCD, ODD, and even eating disorders. The different expression profiles of the isoforms and their significant influence on the excitatory–inhibitory balance in crucial brain structures may explain the phenotypic variability. A thorough analysis of the altered transcripts is mandatory to understand their role in the phenotypic outcome. Further studies on patients carrying *DLG2* alterations will be helpful to enrich clinical description and to elucidate the molecular mechanisms subtended to NDDs.

## Figures and Tables

**Figure 1 genes-13-00859-f001:**
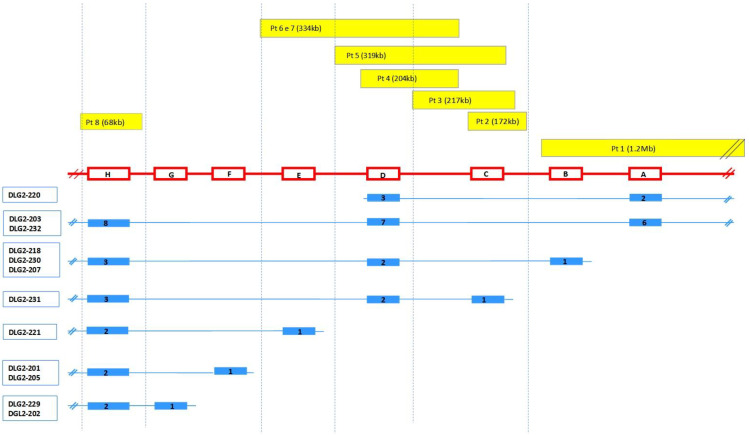
Genetic overview of *DLG2* CNVs and the relative transcripts involved. Top: the yellow bars indicate the imbalances detected for each patient and their extent. Middle: coding regions are indicated by red boxes (right to left) and introns by red lines; A: ENSE00001469378; B: ENSE00002179742, ENSE00002149728, ENSE00002188311; C: ENSE00003840932; D: ENSE00003487678, ENSE00002174027; E: ENSE00002152856; F: ENSE00001532570; G: ENSE00003587836, ENSE00001469384; H: ENSE00003471737. For details, see Appendix A. Bottom: *DLG2* isoforms (blue) with the number of exons relative to each transcript are reported. Relative lengths are not to scale. Pt = patient.

**Figure 2 genes-13-00859-f002:**
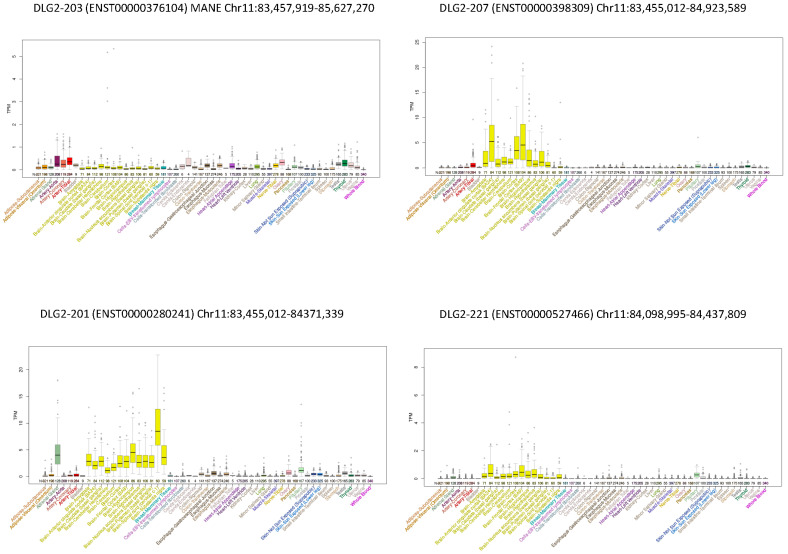
Expression of some DLG2 transcripts across tissues from the GTEx Project. Examples of expression profiles: DLG2-203 (MANE) was ubiquitous; DLG2-207, DLG2-201, and DLG2-221 were prevalently expressed in CNS, with different patterns in brain areas.

**Table 1 genes-13-00859-t001:** Neuropsychiatric features of the patients.

Patient	1	2	3	4	5	6	7	8
Gender	M	M	M	M	F	M	M	M
Age	9 Years	7 Years	8 Years	16 Years	4 Years	15 Years	12 Years	8 Years
**Neuroimaging andneurophysiology**	Brain MRI: Normal EEG: Not Obtained	Brain MRI: Normal EEG: Not Obtained	Brain MRI: Normal EEG: polymorphic slow sharp waves, prevalent on central-posterior areas, tending to diffusion	Brain MRI: Normal EEG: Not Obtained	Brain MRI: Not Obtained EEG: Not Obtained	Brain MRI: Normal EEG: paroxysmal anomalies on the right posterior parietal-occipital-temporal areas and at posterior vertex, sometimes diffusing contralaterally	Brain MRI: thin corpus callosum and enlarged periencephalicfronto-parietal liquoral spaces EEG: Not Obtained	Brain MRI: cavum septi and cavum vergae persistence, enlarged periencephalic subarachnoid spaces, particularly on perifrontal-temporal-polar areas bilaterally EEG: during sleep, rare sharp waves and spikes on the right fronto-central areas; during awake, absence of paroxysmal anomalies
**Neuropsychiatric test**	WISC-IV: TIQ 62, VCI 62, PRI 80, WMI 64, PSI 82 CPRS-R: positive K-SADS-PL	WISC-IV: TIQ 58, VCI 74, PRI 76, PSI 50. CPRS-R: positive K-SADS-PL	WISC-IV: VCI 46, PRI 67 CPRS-R: positive	WISC-IV: TIQ 61, VCI 78, PRI 71, WMI 73, PSI 62	WPPSI-III: TIQ 62 Speech test battery	WISC-IV: TIQ 44, VCI 64, PRI 61, WMI 55, PSI 47 CPRS-R: positive K-SADS PLADOS 2 Module 3: negative CBCL: anxiety-depression, withdrawal	WISC-IV: VCI 60, PRI 41, WMI 58, PSI 47 K-SADS PL ADOS 2 Module 3: Negative CBCL: anxiety-depression, withdrawal, somatic complaints, ADHDY-BOCS: aggressive and contamination obsessions; order, washing and control compulsions	WISC-IV: TIQ 61, VCI 64, PRI 82, WMI 70, PSI 65. ADOS-2 Module 2: positive
**Cognitive-Behavioralphenotype**	Mild ID, ADHD, receptive-expressive LD	Mild ID, ADHD, ODD, mood disorder, anxiety disorder	Mild-moderate ID, ADHD, ODD, verbal and motor dyspraxia, receptive-expressive LD	Mild ID, ADHD, anxiety disorder	Mild ID, motor and verbal dyspraxia	Moderate ID, anxiety-obsessive disorder, binge eating, autistic traits	Moderate ID, ADHD, autistic traits anxiety disorder, focal epilepsy, sleep disturbance, motor clumsiness	Social and cognitive regression after 18 months of age, Mild ID, ASD

ADHD: attention deficit hyperactivity disorder; ADOS 2: Autism Diagnostic Observation Schedule, Second Edition; ASD: autism spectrum disorder; CBCL: Child Behavior Checklist; CPRS-R: Conners Parents Rating Scales-Revised; EEG: electroencephalogram; F: female; ID: intellectual disability; K-SADS PL: Kiddie Schedule for Affective Disorders and Schizophrenia-Present and Lifetime version; LD: language disorder; M: male; MRI: magnetic resonance imaging; OCD: obsessive-compulsive disorder; ODD: oppositional-defiant disorder; PRI: perceptual reasoning index; PSI: processing speed index; TIQ: total intelligence quotient; VCI: verbal comprehension index; WISC-IV: Wechsler Intelligence Scale for Children-IV edition; WMI: working memory index; WPPSI-III: Wechsler Preschool and Primary Scale of Intelligence-III edition; Y–BOCS: Yale–Brown Obsessive Compulsive Scale.

**Table 2 genes-13-00859-t002:** Genetic data about the CNVs detected in the 8 patients.

Pt	Position (Hg38)	Extent	Gene Content	Classification	Transcripts Involved	Inheritance
1	11q14.1 (84, 870, 189–86, 142, 753) × 1	1.27 Mb	*DLG2* *TMEM126B*, *TMEM126A*, *CREBZF*, *CCDC89*, *SYTL2*, *CCDC83*, *PICALM*	LP/P	DLG2-203, DLG2-232 → exons 1-2-3-4-5-6 DLG2-215 → exons 1-2-3-4 (all) DLG2-207, DLG2-218, DLG2-230 → exon 1 DLG2-220 → exons 1–2	Mat
2	11q14.1 (84, 656, 195–84, 828, 622) × 1	172 Kb	*DLG2*	LP/P	DLG2-231 → exon 1	Pat
3	1q42.2 (234, 056, 071–234, 171, 023) × 3	115 Kb	*SLC35F3*	VUS		NA
	5q14.1 (78, 968, 970–79, 263, 190) × 3	294 Kb	*ARSB*, *DMGDH*, *BHMT2*, *JMY*	VUS		NA
	11q14.1 (84, 564, 970–84, 781, 814) × 1	217 Kb	*DLG2*	LP/P	DLG2-231 → exon 1	NA
4	11q14.1 (84, 503, 660–84, 708, 459) × 1	204.8 Kb	*DLG2*	LP/P	DLG2-203, DLG2-232 → exon 7 DLG2-207, DLG2-218, DLG2-230 → exon 2 DLG2-231 → exon 2 DLG2-220 → exon 3	Pat
5	11q14.1 (84, 446, 741–84, 766, 186) × 1	319 kb	*DLG2*	LP/P	DLG2-203, DLG2-232 → exon 7 DLG2-207, DLG2-218, DLG2-230 → exon 2 DLG2-231 → exons 1–2 DLG2-220 → exon 3	Pat
6	11q14.1 (84, 374, 730–84, 708, 459) × 1	334 Kb	*DLG2*	LP/P	DLG2-203, DLG2-232 → exon 7 DLG2-207, DLG2-218, DLG2-230 → exon 2 DLG2-231 → exon 2 DLG2-220 → exon 3 DLG2-221 → exon 1	Mat
	Xp22.31 (7, 666, 592–8, 147, 112) × 3	530 Kb	*VCX*, *PNPLA4*, *MIR561*	VUS		Mat
7	11q14.1 (84, 374, 730–84, 708, 459) × 1	334 Kb	*DLG2*	LP/P	DLG2-203, DLG2-232 → exon 7 DLG2-207, DLG2-218, DLG2-230 → exon 2 DLG2-231 → exon 2 DLG2-220 → exon 3 DLG2-221 → exon 1	NA
8	11q14.1 (84, 187, 699–84, 251, 298) × 3	63.6 Kb	*DLG2*	LP/P	DLG2-203, DLG2-232 → exon 8 DLG2-207, DLG2-218, DLG2-230 → exon 3 DLG2-231 → exon 3 DLG2-221 → exon 2 DLG2-201, DLG2-205 → exon 2 DLG2-229, DLG2-202 → exon 2	Mat

For each patient, the CNVs detected are reported with the position of the first and last abnormal probe (hg38), the extent, the gene content of each CNV, the altered transcripts (Ensembl database), and the inheritance. Pt = patient; P = pathogenic; LP = likely pathogenic; VUS = variant of unknown significance; NA = not available; Mat: maternal; Pat: paternal.

## Data Availability

Publicly available datasets were analyzed in this study. This data can be found at: https://www.deciphergenomics.org/ (accessed on 5 April 2022); https://www.genome.ucsc.edu/ (accessed on 5 April 2022); https://www.genecards.org/ (accessed on 5 April 2022); https://omim.org/ (accessed on 5 April 2022); https://www.ensembl.org (accessed on 5 April 2022); https://www.gtexportal.org/home (accessed on 5 April 2022); http://projects.tcag.ca/variation (accessed on 5 April 2022); http://www.ncbi.nlm.nih.gov/pubmed (accessed on 5 April 2022); and https://www.sigu.net (accessed on 5 April 2022).

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
