# Peer review of "Enhancing DLG2 Implications in Neuropsychiatric Disorders: Analysis of a Cohort of Eight Patients with 11q14.1 Imbalances"

_genes, 2022, doi:10.3390/genes13050859_

Round 1
Reviewer 1 Report
Bertini and colleagues present DLG2 implications in neuropsychiatric disorders The study is of value to the literature. However, this manuscript needs a significant degree of editing (see below).
Introduction
- Why did you select DLG2 gene for analysis? Are there other genes for synapse formations?
Materials and Methods
- “CNVs classified as pathogenic, likely pathogenic or variant of unknown significance (VUS) were reported according to the guidelines of the Italian Society of Human Genetics (https://www.sigu.net), the American College of Medical Genetics guidelines [14], and the European Guidelines for constitu-tional cytogenetic analysis [15].”
=> Did the CNVs classification (Pathogenic, Likely pathogenic, VUS, Likely benign, Benign) of your patients present in your article?
- Did all parents of your patients have array-CGH, too? Did these parents have symptoms about NDD?
Results
- “The 11q14.1 CNVs detected by Array-CGH ranged from 68 Kb to 1.27 Mb; they included one intragenic duplication of DGL2, six intragenic deletions, with two patients 3 sharing an identical genomic imbalance, and one larger deletion harboring additional 4 genes at the 5' of DLG2; two cases presented additional rare CNVs (Table 2).” At page 7.
=> DGL2 should be DLG2.
- 3.3 DGL2 at page 10.
=> DGL2 should be DLG2.
Discussion
- “DLG2 shows incomplete penetrance; alterations of this gene have been reported also in healthy people, and some patients with NDDs inherited the CNVs from an asymptomatic/paucisymptomatic parent, like six cases in our cohort.” at page 15, line 30
=> Are there some references to support your view?
- “Even if in mice models the association between Dlg2 deletion and motor coordination has not been confirmed.” at page 16, line 92.
=> Dlg2 should be DLG2.
Author Response
Introduction
- Why did you select DLG2 gene for analysis? Are there other genes for synapse formations?
Answer:
Genes related to synaptopathies are many (we added new references, n.2, n.3, n.4) and we focused on DLG2 since its role in NDDs has been well investigated in the mouse model, whereas in humans data are still scanty.
Materials and Method
Did the CNVs classification (Pathogenic, Likely pathogenic, VUS, Likely benign, Benign) of your patients present in your article?
We added a column in Table 2 with the CNV classification and modified the legend.
Did all parents of your patients have array-CGH, too? Did these parents have symptoms about NDD?
Parents of Pt.4 and Pt.8 were analyzed by array CGH, whereas in the other parents stusied the CNV inheritance was performed by Real Time PCR. In Table 2 we have reported the parents who performed the segregation analysis, and their family history is reported in Table S1.
Results
=> DGL2 should be DLG2.
- 3.3 DGL2 at page 10.
=> DGL2 should be DLG2.
We made these corrections.
Discussion
- “DLG2 shows incomplete penetrance; alterations of this gene have been reported also in healthy people, and some patients with NDDs inherited the CNVs from an asymptomatic/paucisymptomatic parent, like six cases in our cohort.” at page 15, line 30
=> Are there some references to support your view?
We have added in the text a reference supporting this information (Reggiani et al., 2017);
=> Dlg2 should be DLG2.
We used the lower case since we refer to mouse model.
Reviewer 2 Report
In this paper, Bertini and Milone et al. provide a detailed clinical description of 8 patients with different neurodevelopmental disorders showing 11q14.1 imbalances involving DLG2 gene. Moreover, the authors provide information on the alternative transcripts affected by the identified CNV and their expression in different tissues according to GTEx. Overall, the work is interesting as it highlights the potential involvement of the different isoforms of DLG2 in NDD.
I have some minor concerns and suggestions:
- First of all, by reading this sentence in the abstract “A thorough analysis of the altered transcripts was conducted for each patient.”, it can be expected that the authors analysed the expression of the altered transcripts in the clinical samples, while they only show their expression in different tissues reported in GTEx database. Thus, the authors must specify that it was a bioinformatic analysis. Also in the Discussion, line 43-45, the authors need to explain that they performed a bioinformatic analysis that leads to deduce that all patients have at least one altered transcript highly express in the CNS.
- Please avoid “alternative splicings” when refer to the splicing event or use alternative transcripts if refer to the gene transcripts. For example, in the abstract, please modify this sentence in “with numerous alternative splicings that give rise to isoforms differentially expressed in brain tissues” in “with alternative splicing that give rise to numerous isoforms differentially expressed in brain tissues”.
- To further underline the association of genetic variants in genes involved in synapsis function with NDD, I suggest to cite works such as https://doi.org/10.3389/fncel.2018.00470; doi:10.3390/jcm8020212; doi: 10.1101/cshperspect.a009886.
- In paragraph 3.2, I suggest to indicate the sample IDs in order that the reader can immediately find the information in the Table 2.
- In Table 2, please specify that the transcript IDs are from Ensembl.
- Please check the correspondence between Figure 1 and Supplementary Table 3 (i.e. exon A is exon 2 of the variant 220).
- Please avoid general titles in the Results paragraphs but use titles that reflects the results of the present work (e.g. “Description of the patient cohort”)
- Revise the text considering that there are now changes in the font size and that words in languages different than English must be in italics (i.e. in vitro).
Author Response
- First of all, by reading this sentence in the abstract “A thorough analysis of the altered transcripts was conducted for each patient.”, it can be expected that the authors analysed the expression of the altered transcripts in the clinical samples, while they only show their expression in different tissues reported in GTEx database. Thus, the authors must specify that it was a bioinformatic analysis. Also in the Discussion, line 43-45, the authors need to explain that they performed a bioinformatic analysis that leads to deduce that all patients have at least one altered transcript highly express in the CNS.
We made these corrections in the text, adding 'by bioinformatic analysis', both in the 'abstract' and in the 'discussion'.
(abstract) A thorough bioinformatic analysis of the altered transcripts was conducted for each patient.
(discussion) Since the expression of alternative transcripts is highly tissue-specific [16], we analyzed the expression profiles of the altered transcripts by bioinformatic tools, and, interestingly, all patients have at least one transcript that is highly expressed in the CNS (Figure 2; Figure 1S).
- Please avoid “alternative splicings” when refer to the splicing event or use alternative transcripts if refer to the gene transcripts. For example, in the abstract, please modify this sentence in “with numerous alternative splicings that give rise to isoforms differentially expressed in brain tissues” in “with alternative splicing that give rise to numerous isoforms differentially expressed in brain tissues”.
We made these corrections in the abstract and in the introduction.
- To further underline the association of genetic variants in genes involved in synapsis function with NDD, I suggest to cite works such as https://doi.org/10.3389/fncel.2018.00470; doi:10.3390/jcm8020212; doi: 10.1101/cshperspect.a009886.
We added the suggested references (ref 2,3,4) .
- In paragraph 3.2, I suggest to indicate the sample IDs in order that the reader can immediately find the information in the Table 2.
We have added the Patients' ID in the text, as suggested.
- In Table 2, please specify that the transcript IDs are from Ensembl.
We have made this correction.
- Please check the correspondence between Figure 1 and Supplementary Table 3 (i.e. exon A is exon 2 of the variant 220).
We check the corrispondence and made the correction required.
- Please avoid general titles in the Results paragraphs but use titles that reflects the results of the present work (e.g. “Description of the patient cohort”)
We modified the titles in the Result paragraphs.
Round 2
Reviewer 1 Report
Bertini and colleagues present DLG2 implications in neuropsychiatric disorders The study is of value to the literature. However, this manuscript needs a minor degree of editing (see below).
Materials and Methods
- In Table 2, what is the inherited pattern of DLG2 gene to cause NDDs? Please also supply the inherited pattern of other genes in Table 2.

Author Response
In Table 2, what is the inherited pattern of DLG2 gene to cause NDDs?
Please also supply the inherited pattern of other genes in Table 2.
We are not sure to understand the suggestion, in particular which data are requested.
The inherited pattern of DLG2 is the following (we indicated it in Table 2):
In Pt.1 the alteration, that includes both DLG2 and other genes, is inherited from the mother.
In Pt.3 the parents were not available for the segregation analysis of the three alterations identified.
In pt.6 both the alterations (on chr 11 and on chr X) are inherited from the mother.